# Finite Element Modelling of the Effect of Adhesive Z-Connections on the Swelling of a Laminated Wood Composite

**Mohammad Sadegh Mazloomi [1],\*, Wenchang He [2] and Philip David Evans [2]**

1    FPInnovations, 2665 East Mall, Vancouver, BC V6T 1Z4, Canada
2    Department of Wood Science, University of British Columbia, Vancouver, BC V6T 1Z4, Canada;
     hewenvictory@gmail.com (W.H.); phil.evans@ubc.ca (P.D.E.)
\*    Correspondence: mohammad-sadegh.mazloomi@fpinnovations.ca; Tel.: +1-604-222-5736

**Abstract:** This study used finite element analysis (FEA) to model the effects of adhesive Z-connections on the thickness swelling of laminated wood composites exposed to water. We hypothesized that the area density, diameter, and spatial distribution of adhesive Z-connections will influence the ability of Z-connections to restrain thickness swelling of the composites. We tested this hypothesis by modelling a wood composite in ANSYS FEA software v. 17.0 to explore the effect of moisture on the thickness swelling of the wood composite. The results were compared with those obtained experimentally. We then examined the effect of the area density, size (diam.), and spatial distribution of the adhesive Z-connections on the thickness swelling of wood composites. Our results showed a positive correlation between the number of adhesive Z-connections in the composites and restriction of thickness swelling following 72 h of simulated moisture diffusion. Similarly, increasing the size of adhesive Z-connections also restricted thickness swelling. In contrast, different spatial distributions of Z-connections had little effect on restraining thickness swelling. Our modelling approach opens up opportunities for more complex designs of adhesive Z-connections, and also to examine the effect of wood properties, such as permeability, density, and hygroscopic swelling ratios on the thickness swelling of laminated wood composites.

**Keywords:** wood composites; laminate; Z-connections; finite element modeling; thickness swelling

## 1. Introduction

Wood composites have numerous end uses because they are inexpensive, have many excellent properties, and are mainly derived from a renewable material. However, they also have a number of disadvantages. For example, during their use, wood composites are sometimes exposed to water and, as a result, undergo irreversible thickness swelling [1]. Thickness swelling of wood composites is a serious problem, and is the combination of two components: reversible thickness swelling due to the natural hygroscopic swelling of wood, and irreversible thickness swelling mainly due to the release of compression strains applied to mats of wood flakes during hot pressing [2–4]. Adhesive bonds between flakes counteract the release of compression strains. Therefore, it is important to understand the interaction of wood and adhesive in wood composites. There has been a large amount of research on the distribution of adhesive in some wood composites [5–15], because it affects their dimensional stability and mechanical properties, even though it only makes up 2–14% by weight of the composites [16,17]. Increasing the level of adhesive in wood composites reduces thickness swelling, but this is a costly approach. An alternative approach is to alter the geometry of the adhesive network so it is better able to resist the strains that develop when the composite is exposed to water. For example, the creation of adhesive bonds oriented perpendicular to the X–Y plane, hereafter called adhesive Z-connections, has

been shown to restrain the moisture-induced thickness swelling of a laminated plywood-type composite made from white spruce (*Picea glauca* (Moench) Voss) or yellow cedar (*Xanthocyparis nootkatensis* (D.Don) Farjon & D.K. Harder) veneer [18]. However, there have been no attempts to optimize Z-direction cross-links in wood composites by altering design parameters such as spacing, diameter, and density of Z-direction reinforcement. It is plausible that these design parameters alter the ability of adhesive Z-connections to restrain thickness swelling of wood composites because the same design parameters influence the strengthening effect of Z-reinforcement in carbon-fibre composites [19–21]. However, experimental testing of the effects of spacing, diameter, and density of Z-direction cross-links on the thickness swelling of wood composites would be time-consuming and expensive.

On the other hand, a finite element analysis (FEA) and computer simulation of the effects of adhesive Z-connections on the hygroscopic swelling of composites would make it possible to easily change the aforementioned design parameters and model their effects on thickness swelling. FEA has been used to model the hygroscopic swelling of wood and polymeric materials [22–27]. Hygroscopic swelling of wood is a physical phenomenon that couples moisture absorption and moisture-induced strain energy. Moisture absorption in wood has been intensively studied and has been widely modeled using Fick's second law of moisture diffusion [28,29]. There are some excellent computational models for drying and the development of drying-related defects in solid wood [30–33]. However, limited research has been conducted on finite element modelling of the hygroscopic swelling of wood or wood composites. Therefore, in this study, we developed a coupled hygroscopic–mechanical finite element model for the hygroscopic swelling of wood, based on the approaches used to model similar phenomena in polymeric materials, but taking into account the directional properties of wood. The ANSYS Multiphysics FEA program was used to simulate moisture diffusion and hygroscopic swelling of a model composite, and the model was verified experimentally. We hypothesized that the area density, diameter, and spatial distribution of adhesive Z-connections will influence the ability of Z-connections to restrain thickness swelling of laminated wood composites. We tested this hypothesis by modelling the thickness swelling of wood composites containing different numbers, sizes, and spatial distributions of adhesive Z-connections. The final output of our simplified model is improved design parameters for laminated plywood-type composites containing adhesive Z-connections.

## 2. Materials and Methods

### 2.1. Experimental Setup

Twenty rotary-cut (tangential) white-spruce veneer sheets measuring 15 cm (width) × 225 cm (length) were purchased from a local retailer of commercial veneer (ENE Wood Products, Surrey, BC, Canada). Six sheets were selected and stored in a conditioned environment (20 ± 1 °C and 65 ± 5% relative humidity) for one month. Samples measuring 25 mm × 25 mm square were cut from each of the six conditioned sheets using a paper cutter (Boston$^{TM}$ 2586). A veneer was selected at random, and fourteen veneer squares were selected at random from the first veneer sheet and allocated to two different types of model composites, each consisting of seven veneer squares: (1) composite composed of perforated veneer squares; (2) composite composed of unperforated veneer squares (control). Veneer squares were perforated with a hand-held high-speed dental drill (W&H$^{®}$ Trend WD-56, Edmonton, Canada) with a 1 mm diameter drill bit (Dentsply$^{®}$ TN burr, Charlotte, NC, USA) operating at 6000 rpm with 100 g/cm of torque.

The drill produced accurately sized, smooth, cylindrical, 1 mm diameter holes in the veneer. Seven veneer squares were placed in a mould to align them vertically and all seven were perforated together in the mould (Figure 1). The spacing between holes was 5.5 mm. All the perforated and unperforated veneer squares were stored in a conditioned environment (as above) for seven days.

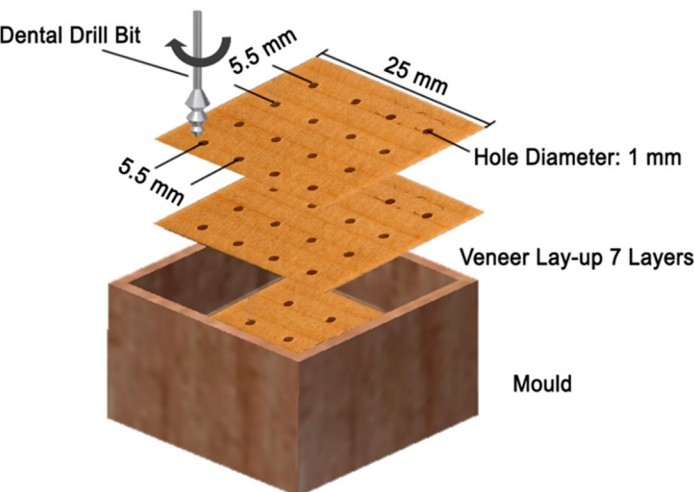

**Figure 1.** Schematic illustration of veneer lay-up and hole pattern.

Two matching groups of perforated or unperforated veneer squares cut from the same parent veneer sheet were selected, and 0.044 g ± 0.0005 g (40 µL) of a polyurethane adhesive (Gorilla glue®) was applied to each veneer using a syringe, and evenly spread across the surface of each veneer square using a glass coverslip. The same amount of adhesive was applied to perforated and unperforated veneers. Veneers were laid up with their grain direction parallel to each other to make model composites containing either perforated or unperforated veneer squares. Composites were placed in a small laboratory press and pressed at room temperature at 13.8 MPa for eight minutes. As a result of applying pressure to veneers, adhesive flows into the vertical holes and creates adhesive Z-connections. The resulting laminated composite specimens were conditioned at 20 ± 1 °C and 65 ± 5% r.h. for three days, and the thickness of each specimen was measured in 12 places using a digital caliper (Mitutoyo®). Composite specimens from the first veneer sheet for each species were submerged in water to a depth of 1 cm for 24 h, in a tank measuring 15 cm × 8 cm × 6 cm, and their thicknesses were remeasured, as above.

Finally, the changes in the surface topography of the specimens after immersion in water and drying was measured using a confocal profilometer (Altisurf 500®, Marin, France). The model composite was placed on the stage of a confocal profilometer, and the height of the entire surface of the specimen was measured with a 3 mm probe at a scanning speed of 6 mm/s. The spacing between measurement points was 10 µm × 10 µm (x- and y-directions) and the accuracy of measurements in the Z-direction was 0.00033 µm, according to the manufacturer of the profilometer. Height information from all the scans was imported into image-analysis software (AltiMap® Premium v. 6.2.) on a PC. This software was used to obtain topographical information on the model wood composites including the difference in swelling of the wood and Z-connections (adhesive plugs), and also to produce colour-coded height maps of the surface of the composites.

### 2.2. Moisture Diffusion Modeling

Moisture diffusion in wood is governed by Fick's second law, which can be expressed using Equation (1) [25,27]:

$$\frac{\partial C}{\partial t} = \left\{ D_x \frac{\partial^2 C}{\partial x^2}, D_y \frac{\partial^2 C}{\partial y^2}, D_z \frac{\partial^2 C}{\partial z^2} \right\} \tag{1}$$

where $C$ is the local moisture concentration (kg/m³), t is time (s), $D_x$, $D_y$, and $D_z$ represent the coefficient of the moisture diffusivity of wood in the X (longitudinal), Y (tangential) and Z (radial) directions, respectively. Most commercial FEA software, with the exception of COMSOL software, does not have a moisture-diffusion simulation module. However, there is a similarity between the governing equations for moisture diffusion and ther-

mal diffusion [34]. Thermal diffusion (transient heat conduction) can be described using Equation (2) [25–27].

$$\frac{\partial T}{\partial t} = \left\{ \alpha_{Tx} \frac{\partial^2 T}{\partial x^2}, \alpha_{Ty} \frac{\partial^2 T}{\partial y^2}, \alpha_{Tz} \frac{\partial^2 T}{\partial z^2} \right\} \tag{2}$$

where $T$ is local temperature (°C), t is time (s), and $\alpha_{Tx}$ is thermal diffusivity (m$^2$/s) in X-direction, and can be defined as $\alpha_{Tx} = K_x/(\rho C p)$ where $K_x$ is thermal conductivity in X-direction (W/m°C), ρ is density (kg/m$^3$), and $C_p$ is specific heat (J/kg°C).

The similarity between the two diffusion equations makes it possible to use the thermal diffusion simulation module in the FEA software ANSYS v. 17.0 to solve moisture-diffusion problems [24]. However, the software is only directly applicable if a homogeneous system is modelled. For heterogeneous composite systems, the moisture concentration is discontinuous at the multi-material interface [34–37]. This discontinuity is a barrier to the use of FEA's thermal-diffusion module to solve moisture diffusion problems [38]. But this barrier can be circumvented by introducing a new physical quantity, 'wetness', and modelling the moisture diffusion of a composite consisting of two materials, i.e., wood and adhesive. The derivative variable 'wetness', $W$ is defined in Equation (3) [35]:

$$W = \frac{C}{C_{sat}} \tag{3}$$

where $C_{sat}$ is the saturated moisture concentration (kg/m$^3$) of a material. The variable 'wetness', $W$ is continuous across a bi-material interface and, therefore, obeys Fick's law of diffusion. So, Equation (1) can be re-written as Equation (4) [24,25]:

$$\frac{\partial W}{\partial t} = \left\{ D_x \frac{\partial^2 W}{\partial x^2}, D_y \frac{\partial^2 W}{\partial y^2}, D_z \frac{\partial^2 W}{\partial z^2} \right\} \tag{4}$$

where wetness, $W$, has a physical meaning, in that $W = 0$ indicates a completely dry state, and $W = 1$ indicates a fully saturated state in which both cell walls and cell cavities are saturated with water.

Table 1 compares thermal diffusion and moisture diffusion including the newly introduced physical quantity, wetness, which was used within the ANSYS Multiphysics module to simulate moisture diffusion within model wood composites containing various configurations of adhesive Z-connections (area density, diameter, and spatial distribution).

**Table 1.** Parameters used to model thermal and moisture diffusion.

| Properties | Thermal Diffusion Parameters | Moisture Diffusion Parameters |
|---|---|---|
| Field Variable | Temperature, $T$ | Wetness, $W$ |
| Density | $P$ | 1 |
| Conductivity | $K$ | $D.C_{sat}$ |
| Specific Capacity | $C_p$ | $C_{sat}$ |
| Coefficient of Expansion | $A$ | $\beta.C_{sat}$ |

On the other hand, hygroscopic strain, $\varepsilon h$, induced by moisture diffusion into the material, can be defined using Equation (5), in which $\beta x$, $\beta y$, and $\beta z$ are coefficients of moisture expansion in X-, Y-, and Z-directions, respectively.

$$\left\{ \varepsilon_{hx}, \varepsilon_{hy}, \varepsilon_{hz} \right\} = \left\{ \beta_x M, \beta_y M, \beta_z M \right\} \tag{5}$$

Saturated moisture concentration, $C_{sat}$ of a material can be calculated using Equation (6) [25]:

$$C_{sat} = \frac{M_{sat}}{abc(100 - Vol\%\,of\,Moisture)} \tag{6}$$

where $M_{sat}$ is saturated mass (kg) and *a*, *b*, and *c* are length (m), width (m), and thickness (m) of specimens, respectively.

Wood has a critical moisture-content level, the fibre saturation point (FSP), above which its dimensions will not increase with increasing moisture content. Therefore, the saturated moisture concentration of the model composite derived from Equation (6) is equal to the FSP of wood.

Table 2 shows the mechanical properties of spruce (*Picea* spp.) wood [39] and polyurethane adhesive [40]. Table 3 shows the moisture-diffusion and hygroscopic-swelling properties of spruce (*Picea* spp.) wood. Moisture-diffusion coefficients, *D*, and coefficients of moisture expansion, *β*, of spruce (*Picea* spp.) wood were taken from the literature [41–43], respectively. Moisture-diffusion coefficients, *D*, and coefficients of moisture expansion, *β*, for polyurethane adhesive were also taken from the literature [44–46].

**Table 2.** Mechanical properties of spruce wood and polyurethane adhesive.

| Material | Thermal Elasticity (MPa) | | | Poisson's Ratio | | | Thermal Shear Modulus (MPa) | | |
|---|---|---|---|---|---|---|---|---|---|
| | $E_x$ | $E_y$ | $E_z$ | $v_{xy}$ | $v_{yz}$ | $v_{xz}$ | $G_{xy}$ | $G_{yz}$ | $G_{xz}$ |
| Spruce wood | 10,800 | 842 | 464 | 0.47 | 0.25 | 0.37 | 659 | 32 | 691 |
| Polyurethane adhesive | 760 | | | 0.35 | | | 280 | | |

**Table 3.** Moisture diffusion and hygroscopic swelling properties of spruce wood and polyurethane adhesive [44–46].

| Material | Saturated Moisture Concentration | Moisture Diffusion Coefficient (m²/s) | | | Coefficients of Moisture Expansion | | |
|---|---|---|---|---|---|---|---|
| | $C_{sat}$ (kg/m³) | $D_x$ | $D_y$ | $D_z$ | $β_x$ | $β_y$ | $β_z$ |
| Spruce wood | 143 | $1000 \times 10^{-12}$ | $52 \times 10^{-12}$ | $52 \times 10^{-12}$ | 0.001 | 0.075 | 0.308 |
| Polyurethane adhesive | 20 | | $2 \times 10^{-12}$ | | | 0.001 | |

The hygro–mechanical properties of spruce wood and polyurethane adhesive as input values in the ANSYS module are listed in Table 3.

$E_x$, $E_y$, and $E_z$ = elastic moduli in x, y, and z directions, respectively; $G_{xy}$ = longitudinal shear modulus; $G_{xz}$ and $G_{yz}$ = transverse shear moduli; $v_{xy}$ = longitudinal Poisson's ratio; $v_{xz}$ and $v_{yz}$ = transverse Poisson's ratios.

### 2.3. Finite Element Modeling

One-eighth of the wood composite was modelled using finite element analysis because of the presence of three symmetrical planes, as arrowed in Figure 2. The use of symmetrical planes can significantly reduce the number of elements needed, thereby reducing computational time [47–52]. The definition of the coordinate system used in the finite element model is shown in Figure 2. The X and Y directions represent grain and cross-grain directions of wood veneer, respectively. Z is the thickness direction of wood veneers. The entire wood composite is submerged in water to simulate the water-absorption phenomenon.

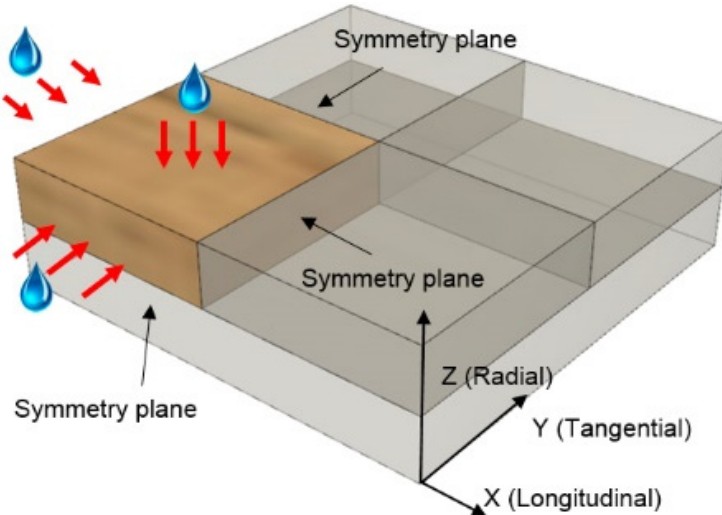

**Figure 2.** Definition of X–Y–Z coordinate system and schematic illustration of the symmetry of the modelled material (arrowed blue) and water absorption planes (arrowed red) in the model.

The initial condition of the entire domain was W = 0 to simulate the originalstatus of a fully unsaturated wood composite. The boundary condition of the three symmetry planes was a zero flux of moisture, and the boundary condition of the three exposed surfaces was W = 1 to simulate the environment surrounding the wood composite. This approach introduced an initial moisture gradient between the surface and the core of the composite and hence allowed moisture diffusion to occur over time. Hexahedron elements (8-noded) with an element size of 0.33 mm were selected after a sensitivity analysis was performed to optimize element size and type on the thickness swelling of the wood composite. The mesh at the wood-adhesive interface was further refined using a contact-area refinement function to ensure a smooth transition between the interface and the bulk of the composite. The latter approach was used because an efficient mesh with a smooth transition at bi-material interfaces usually improves the computation efficiency of finite element models [53].

Moisture diffusion simulation was carried out using the 'transient thermal' module in ANSYS® which produced moisture-concentration profiles in the model wood composites. The hygroscopic-swelling simulation was carried out using the 'transient structural' module. This module was used to examine moisture-induced swelling and Z-direction deformation of the model wood composites. All modelling using ANSYS Multiphysics Workbench software v. 17.0 was performed on a high-end desktop computer (3.40 GHz Intel@ Core i7 2600K processor, 24 GB of random access memory (RAM), 1 TB of hard disk drive, Nvidia@ GeForce GTX 590 graphics card). Average thickness swelling of the top surface of the model was calculated to compare the effects of different design configurations.

## 3. Results

### 3.1. Simulation of Moisture Diffusion

The simulated moisture content of the wood composite changed over time. Figure 3 shows these changes at the core of the model wood composite. Moisture change occurred at a faster rate when the simulation began and slowed down after 24 h. Note in the Figure that a blue colour indicates W = 0 (completely dry); a red colour indicates W = 1 (completely saturated). Hence, it can be observed that the core of the model wood composite was the last to become saturated with water. It is also clear that the moisture-diffusion rate in wood was dependent on diffusion direction. Moisture diffusion was fastest in the X-direction which is aligned with the longitudinal direction of the wood. As a result of such orthotropic moisture diffusion, moisture distribution within the model composite at any given time was not uniform. The simulation also suggested that adhesive Z-connections were not as permeable as wood, as indicated by the blue colour of adhesive Z-connections throughout the water-soaking process. Note that in Figure 3, the minimum wetness in the core of the

model wood composite, which is in the top left corner, is 76%, 96% and 99% for 24, 48, and 72 h of moisture-ingress simulation, respectively. After 72 h (3 days) of simulated water diffusion, wetness of the core of the centremost wood veneer and centremost adhesive Z-connection were 99.3% and 98.1%, respectively (Figure 3), indicating near-complete saturation. As a result, the simulation of water diffusion into the model composite was stopped at 72 h.

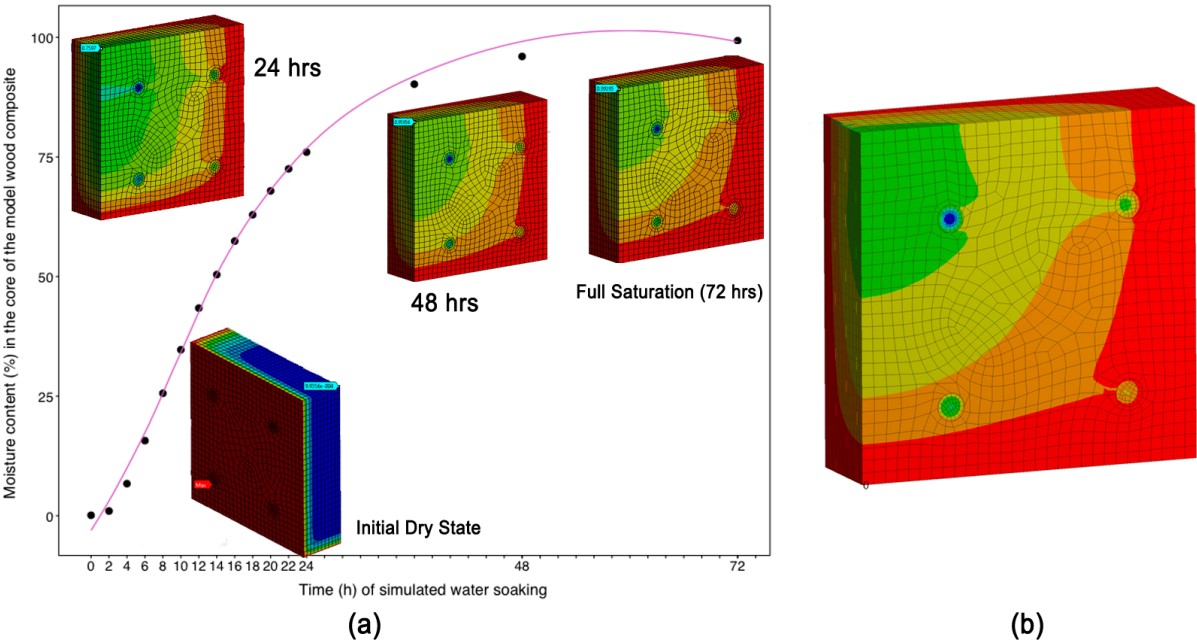

**Figure 3.** (**a**) Simulated-moisture-content change over time in the core of the model wood composite. (**b**) 3D moisture distribution in 1/8 of the model wood composite after 72 h (3 days) of simulated water soaking. W = 0 (completely dry), a red colour indicates W = 1 (completely saturated).

## 3.2. Experimental Verification

Simulated thickness swelling was compared with those obtained from experiments. Thickness swelling after 24 h of moisture-diffusion simulation was averaged over the surface to be verified by the experimental results. The results are shown in Table 4. The simulation results accord with experimental data, which shows that the FE prediction approximates to the measured thickness swelling. The simulation results underestimate experimental measurements as the model does not take into account the recovery of compressive strain imparted during the pressing of the composite. Moreover, the simulated thickness swelling is compared with the result obtained from confocal profilometer scans of the surface. Figure 4 indicates that FE-simulated thickness swelling is similar to the thickness swelling obtained experimentally using a confocal profilometer. In both cases, the surface is not flat. The thickness swelling at the edges is higher, and the adhesive Z-connections caused local restriction of thickness swelling which resulted in the creation of dimples at the surface of the wood composite.

**Table 4.** FE simulation and experimental thickness swelling (TS) of composites with and without adhesive Z-connections after 24 h of water soaking.

| Properties | FE TS * (mm) | Experimental TS (mm) |
|:---:|:---:|:---:|
| Control | 1.78 | 2.25 |
| Z-Connections | 1.40 | 1.74 |

\* The FE simulation of thickness swelling (TS) in this table is based on an initial environmental condition of $20 \pm 1$ °C and $65 \pm 5\%$ r.h. and not completely dry conditions.

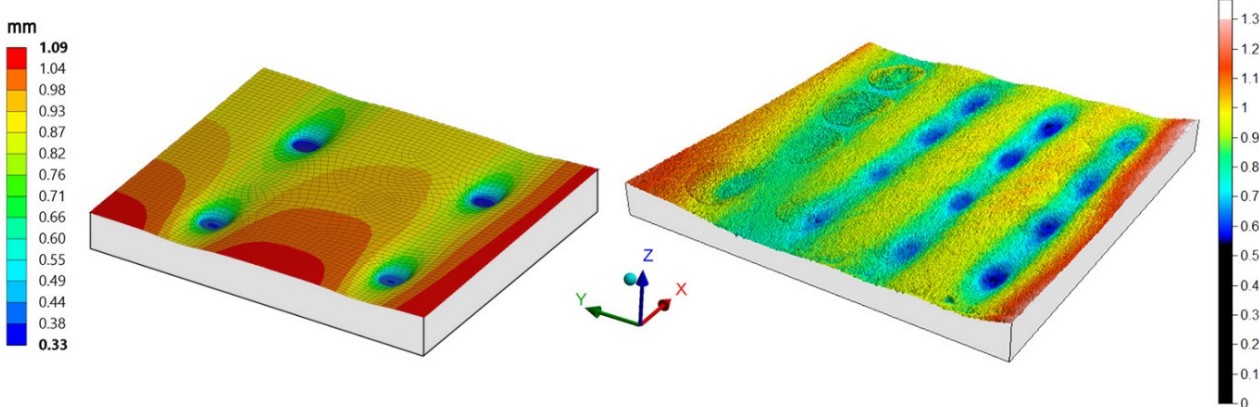

**Figure 4.** Simulated thickness swelling after 24 h, and confocal profilometry scans of the surface after 24 h of immersion in water. Note that the FE model is 1/8 of the model wood composite.

### 3.3. Effects of Area Density, Diameter, and Spatial Distribution of Adhesive Z-Connections on Thickness Swelling of the Model Wood Composite

After model verification, three main parameters related to the design of adhesive Z-connections were examined, i.e., area density, diameter, and the spatial distribution of adhesive Z-connections.

### 3.3.1. Area Density of Adhesive Z-Connections

To see the effect of the area density of adhesive Z-connections on the average thickness swelling of wood composites, five different design configurations with different numbers of adhesive Z-connections were employed. The numbers of adhesive Z-connections in these different wood composites were 4, 16, 20, 36, and 48 and the diameters of adhesive Z-connections were 1 mm.

Table 5 shows the thickness swelling of model wood composites with different Z-connection area densities after 24 and 72 h simulated moisture diffusion. Table 5 shows that the closer adhesive Z-connections were to each other, the better they were at restraining the swelling of the composites. For example, the restriction in thickness swelling was more pronounced in the wood composite with 48 adhesive Z-connections. This composite showed 1.61 mm of thickness swelling after 24 h and 1.74 mm when it was fully saturated. Figure 5a–e shows the simulated thickness swelling after 24 h. It confirms what is readily apparent in Table 5, that an increase in the number of adhesive Z-connections reduces the average thickness swelling of the wood composite. It also shows that 'dimples' created with Z-connections were better connected in the X-direction than in the Y-direction. The result clearly shows a positive correlation between the area density of adhesive Z-connections and their ability to restrain thickness swelling. However, an increased number of adhesive Z-connections was unable to restrain the maximum thickness swelling that occurred at the edges of the composite.

**Table 5.** Average and maximum thickness swelling (TS) of wood composites after 24 and 72 h of simulated moisture diffusion.

| Properties | Average TS (24 h) (mm) | Maximum TS (24 h) (mm) | Average TS (72 h) (mm) | Maximum TS (72 h) (mm) |
|---|---|---|---|---|
| 0 Z-Connections | 2.11 | 2.16 | 2.16 | 2.16 |
| 4 Z-Connections | 1.97 | 2.17 | 2.10 | 2.16 |
| 16 Z-Connections | 1.86 | 2.19 | 2.01 | 2.19 |
| 20 Z-Connections | 1.83 | 2.20 | 1.98 | 2.20 |
| 36 Z-Connections | 1.71 | 2.22 | 1.84 | 2.21 |
| 48 Z-Connections | 1.61 | 2.22 | 1.74 | 2.18 |

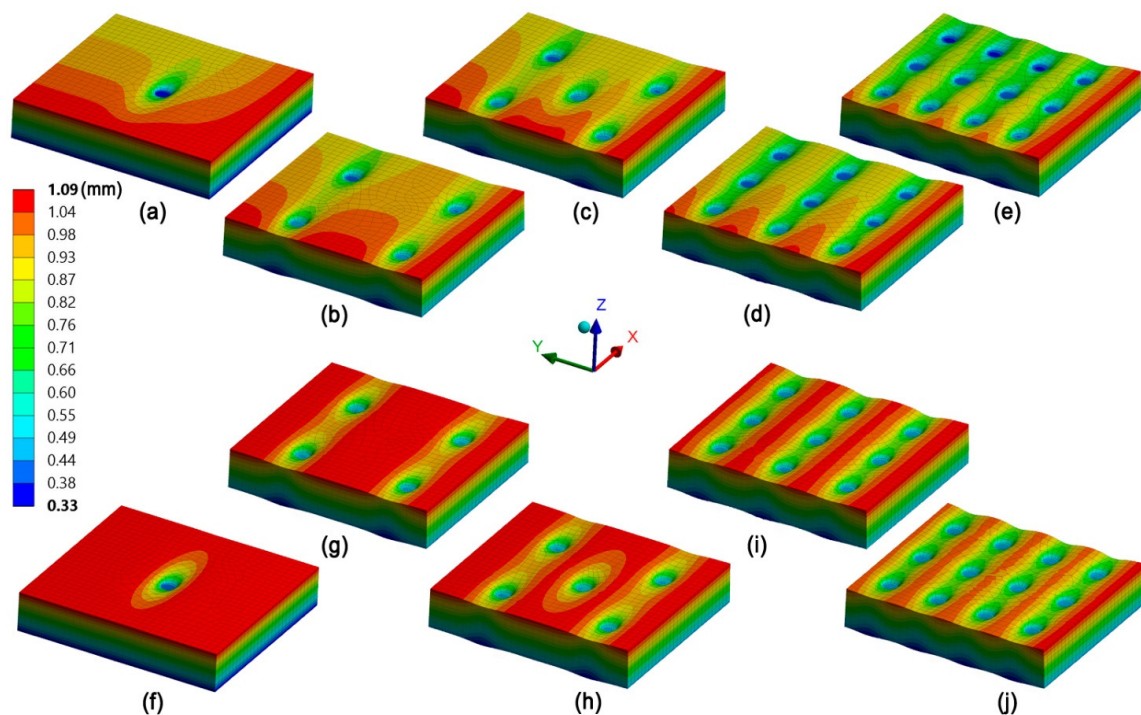

**Figure 5.** Z-direction deformation in 1/8 of the model wood composites containing 4, 16, 20, 36, and 48 adhesive Z-connections; (**a**–**e**) after 24 h simulated immersion in water; (**f**–**j**) after 72 h simulated immersion in water. Note that the maximum in the Figure is 1.09 as it is 1/8 of the model wood composite. This number should be doubled in order to reach the correct thickness swelling.

### 3.3.2. Diameter of Adhesive Z-Connections

The effects of the diameter of adhesive Z-connections on the thickness swelling of the model wood composites were also investigated. Three wood composites were modeled with 16 adhesive Z-connections with diameters of 1, 1.5, and 2 mm (Figure 6). As can be seen in Table 6, the size of the 'dimples' around the adhesive Z-connections were positively correlated with the diameter of the Z-connections, as expected. The maximum thickness swelling of the wood composite was 2.19 mm for all three designs after 72 h of simulated moisture diffusion.

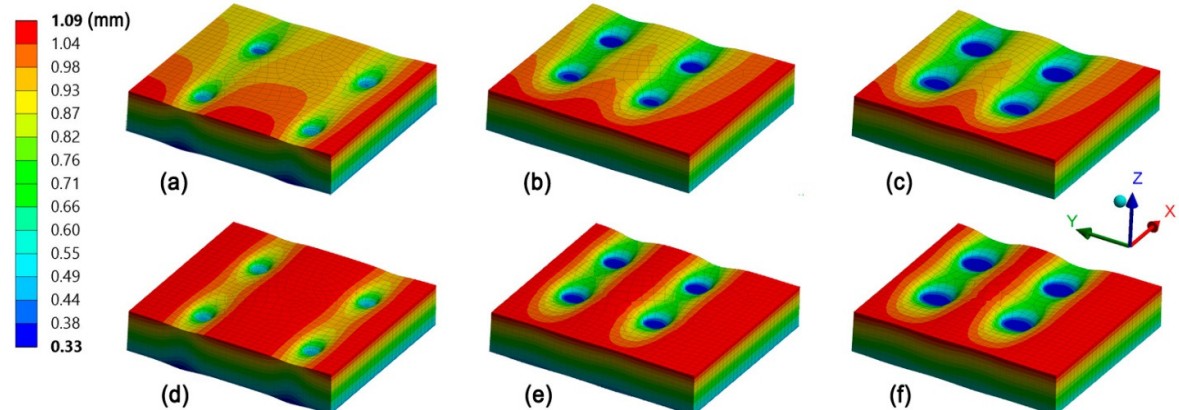

**Figure 6.** Z-direction deformation in 1/8 of model wood composite containing simulated adhesive Z-connections with different diameters; (**a**–**c**) after 24 h simulated immersion in water; (**d**–**f**) after 72 h simulated immersion in water.

**Table 6.** Average and maximum thickness swelling (TS) of wood composites with different adhesive diameters.

| Properties | Avg. TS (24 h) (mm) | Max. TS (24'h) (mm) | Avg. TS (72 h) (mm) | Max. TS (72 h) (mm) |
|---|---|---|---|---|
| $\Phi$ 1 mm Z-Connections | 1.86 | 2.19 | 2.01 | 2.19 |
| $\Phi$ 1.5 mm Z-Connections | 1.77 | 2.18 | 1.90 | 2.19 |
| $\Phi$ 2 mm Z-Connections | 1.65 | 2.20 | 1.77 | 2.19 |

3.3.3. Spatial Distribution of Adhesive Z-Connections

Thickness swelling of the composite containing simulated adhesive Z-connections with different spatial distributions but identical area densities are shown in Figure 7. Two different distributions were modelled: (1) square (Figure 7a,c,e,g), and (2) diagonal (Figure 7b,d,f,h). The average and maximum thickness swelling for these configurations are shown in Table 7. The table suggests that for both 16 Z-connections and 20 Z-connections, the spatial distribution did not have any effect on the average thickness swelling of the wood composite after 24 and 72 h of simulated moisture diffusion. The average thickness swelling for both the square and diagonal arrangements of adhesive in wood composites with 16 Z-connections was 1.86 and 2.01 mm for 24 and 72 h of moisture diffusion, respectively. The average thickness swelling after 72 h for both arrangements of Z-connections in wood composites with 20 Z-connections was 1.98 mm.

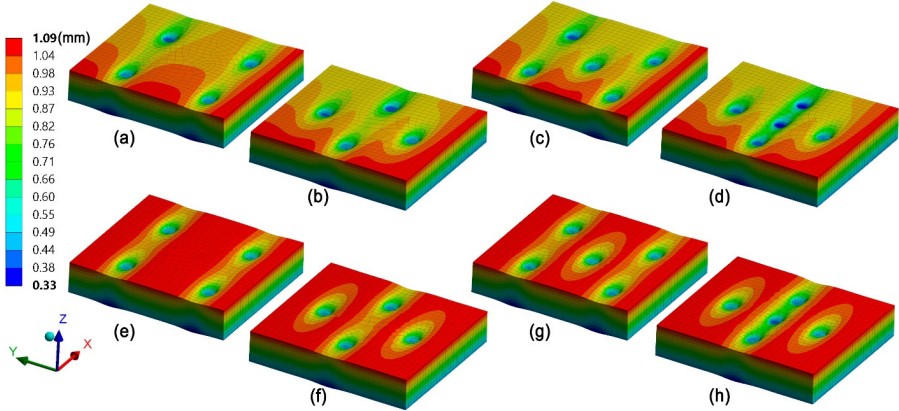

**Figure 7.** Z-direction deformation in 1/8 of a model wood composite containing 16 and 20 simulated adhesive Z-connections with different spatial distributions; (**a**–**d**) after 24 h simulated immersion in water; (**e**–**h**) after 72 h simulated immersion in water.

**Table 7.** Comparison of the effects of the square and diagonal spatial distribution of simulated adhesive Z-connections on thickness swelling (TS) of model wood composites.

| Properties | Avg. TS (24 h) (mm) | Max. TS (24 h) (mm) | Avg. TS (72 h) (mm) | Max. TS (72 h) (mm) |
|---|---|---|---|---|
| 16 Z-Connections, Square Layout | 1.86 | 2.19 | 2.01 | 2.19 |
| 16 Z-Connections, Diagonal Layout | 1.86 | 2.19 | 2.01 | 2.19 |
| 20 Z-Connections, Square Layout | 1.83 | 2.20 | 1.98 | 2.20 |
| 20 Z-Connections, Diagonal Layout | 1.84 | 2.20 | 1.98 | 2.20 |

## 4. Discussion

The area density and diameter of simulated adhesive Z-connections influenced the ability of the adhesive Z-connections to restrain the thickness swelling of model wood composite as hypothesized, while the spatial distribution of adhesive Z-connections did not have any effect on the thickness swelling of the model wood composite.

As expected, a larger number of adhesive Z-connections over a fixed area was better at restraining the thickness swelling of model wood composites than a lower number of adhesive Z-connections. There was a linear correlation between the thickness swelling of wood (excluding the thickness-swelling components of adhesive Z-connections) and the area occupied by adhesive Z-connections (Table 5). This correlation accords with a study by Partridge and Cartié [21]. They examined the effects of three different area densities (0.5%, 2%, and 4%) of Z-pin insertions on the fracture toughness of a carbon fibre/epoxy prepreg composite. They found that the load-carrying capability and Mode I fracture toughness of the composite were positively correlated with the area density of Z-pins.

A related approach to increasing the area density of adhesive Z-connections is to increase the diameter of individual Z-connections. Simulation results suggested that adhesive Z-connections with larger diameters were better than smaller diameter ones at restraining thickness swelling of the model wood composites. This strengthens the correlation between the area density of adhesive Z-connections and the reductions in thickness swelling of the model wood composites mentioned above. Simulation results clearly suggested that increasing the area density of adhesive Z-connections better restrained thickness swelling of the model composites. But, from a practical point of view, it would not be economical to simply increase the area density of adhesive Z-connections because adhesive is expensive, and, accordingly, composites containing large numbers of adhesive Z-connections would be too expensive. A more practical approach is to better distribute adhesive Z-connections to reduce thickness swelling. On this topic, one interesting finding from this study is that having more adhesive Z-connections with a smaller diameter is better at restraining the thickness swelling of wood composites than a lower number of larger-diameter adhesive Z-connections. The area density of a wood composite with 16, 1.5 mm adhesive Z-connections is the same as that of a composite with 36, 1 mm diameter adhesive Z-connections. However, our results show that the latter is better at restraining thickness swelling after 24 and 72 h of moisture diffusion. Moreover, the area density of Z-connections in the wood composite with 48, 1 mm diameter adhesive Z-connections is less than that of the composite with 16, 2 mm diameter adhesive Z-connections. However, the former configuration was better at restricting thickness swelling. These positive effects occurred because the additional Z-connections were able to bridge the four Z-connections, creating a more interconnected network of 'dimples'. This improved array of Z-connections was better at restraining the swelling of the composites.

In summary, using more adhesive Z-connections with smaller diameters appears to be the most efficient approach for improving the ability of adhesive Z-connections to more evenly reduce thickness swelling of the model laminated wood composites. Such an approach does not increase the area density of adhesive Z-connections needed in the model wood composites; therefore, a similar or reduced amount of adhesive can be used to restrict thickness swelling.

## 5. Conclusions

In this study, we developed a finite element model (FEM) coupling moisture diffusion and hygroscopic swelling in a model laminated plywood-type composite containing adhesive Z-connections. Simulation results, using the ANSYS® FEA package, partially supported our 'specific hypothesis' that design parameters such as area density and diameter of Z-connections influence the thickness swelling of the model wood composites, while their spatial arrangement does not have a significant effect on restricting thickness swelling. In general, as expected, there was a positive relationship between numbers of Z-connections (area density) and the ability of the Z-connections to restrain thickness swelling. More importantly, from a practical point of view, at the same area density, a higher number of Z-connections with a smaller diameter was better than a lower number of Z-connections with a larger diameter.

In conclusion, finite element analysis (FEA) is clearly a useful tool for exploring the effects of various design parameters affecting the ability of adhesive Z-connections to reduce



thickness swelling of wood composites. FEA provided useful information on the effects of area density, diameter, and spatial arrangement of adhesive Z-connections on the thickness swelling of model wood composites, as mentioned above. Our modelling approach also opens up opportunities for more complex designs of adhesive Z-connections. It would be worthwhile, for example, to conduct further research on using the FEA model to examine effects of wood properties, such as permeability, density, and radial/tangential-swelling ratio on the thickness swelling of the model wood composites.

**Author Contributions:** M.S.M. performed all of the modeling work. W.H. performed the experimental work. M.S.M. and P.D.E. wrote the first draft of the manuscript, and all three authors discussed and commented on the results and contributed to the final version of the manuscript. All authors have read and agreed to the published version of the manuscript.

**Funding:** This research received no external funding.

**Informed Consent Statement:** Not applicable.

**Data Availability Statement:** The data presented in this study are available on request from the corresponding author.

**Acknowledgments:** Wenchang He thanks the China Scholarship Council for a scholarship. P.D.E. thanks Viance, Tolko, FPInnovations, the Faculty of Forestry at UBC and the Government of British Columbia for their support of his BC Leadership in Advanced Forest Products Manufacturing Technology at the University of British Columbia (UBC). M.S.M. and P.D.E. thank the Faculty of Forestry at UBC for their financial support of this work, which is part of a program of research for P.D.E.'s BC Leadership.

**Conflicts of Interest:** The authors declare that they have no affiliations to or involvement with organizations that have financial interests in the subject matter or materials discussed in this paper.

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
