# Peer review of "Finite Element Modelling of the Effect of Adhesive Z-Connections on the Swelling of a Laminated Wood Composite"

_jcs, doi:10.3390/jcs7100442_

Round 1

Reviewer 1 Report

In the present work, the authors developed a computational model, using a commercial finite element program ANSYS, to study the moisture induced swelling behavior of laminated wood composites reinforced with the help of adhesive z-connections. The authors used experimental results to validate their computational model. Following this, they used the developed model to study certain design parameters, such as the z-connection areal density, diameter and the spatial arrangement. The reviewer feels that the authors have spent considerable effort in developing the methodology for this work. All results are presented clearly. Most conclusions are reasonable. The presentation of results can be improved in certain areas. These are noted in the points below.

1.       Line 53: Have the authors done any exploratory work on the time/material resources needed to conduct this study? Are there relevant experimental works in the literature studying the effect of Z-direction reinforcement to mitigate swelling of laminated wood composites?

2.       Line 144: The authors state that W is continuous across a bi-material interface. Is this an assumption or is this an established model studied previously?

3.       Line 202: Can the authors comment further on the nature of the sensitivity analysis performed?

4.       Figure 3: Can contour levels be added next to the contours shown in figure 3? It becomes hard to understand it without the contour levels.

5.       Line 272-300: This is impressive work. However, the reviewer strongly suggests summarizing this paragraph and presenting the results only with a table. Another useful addition to the paper would be an inclusion of plots showing the variation of average and maximum thickness, both 24h and 72h, versus the number of z-connections. That would provide visual confirmation of the conclusions being discussed in this text. One more comment – is there a reason for the staggered presentation of the images in figure 5? Is this to save space? It would be more straightforward to organize them in rows and columns rather than this approach.

6.       Line 314-325: Same comment as above.

7.       Line 379: Can the authors provide the exact comparison that they are talking about here in a tabular form? This is interesting for a reader to note.

Author Response

Thanks very much for your thorough review and your comments.
Please see our response attached. 

Reviewer 2 Report

The present paper presents a comprehensive Finite Element Analysis (FEA) model investigating the Influence of Adhesive Z-Connections on Hygroscopic Swelling in Laminated Wood Composites. While the study offers valuable insights, several essential considerations should be addressed before finalizing the publication.

The complexity of moisture movement modeling, a multifaceted phenomenon, warrants acknowledgment of existing literature. Numerous pertinent papers, including those addressing moisture movement during drying, have been overlooked in the citation. The adopted model simplifies moisture transport to diffusion-based mechanisms, with diffusion coefficients (D) as the primary determinants. However, the methodology behind obtaining diffusion coefficient values for various directions and adhesive types remains absent. Furthermore, a clarification of the standard definition of moisture saturation is advised, as conventionally, "fully saturated" denotes reaching maximum moisture capacity rather than the fiber saturation point.

A fundamental concern pertains to the initial condition of W=0 signifying complete dryness. Inaccuracies arise if the model assumes absolute dryness at the outset, as wood materials inherently retain some moisture contingent on environmental conditions until subjected to oven drying. To enhance validity, it is crucial to validate the moisture content predictions against experimental data, given the interdependence of hygroscopic swelling and moisture content. A thorough elucidation of the hygroscopic swelling model is warranted to ensure comprehensibility.

Within the introduction, clarifying whether the study focuses on a specific wood composite, such as Laminated Veneer Lumber (LVL) or Plywood, is recommended. Similarly, defining "Z-connection" concisely would facilitate understanding for readers unfamiliar with the term.

Is 13.8 MPA pressure too high for manufacturing LVL? What the recommended pressure?

Line 128 mentions most FEA software does not have a moisture diffusion model – this statement is wrong- e.g. COMSOL has.

Table 3 – add similar title above D, and betas describing what they are (make it similar to Table 2).

Line 187 could be more effectively worded as "…resulting in reduced computational time."

Line 196 and line 223 W=0- fully dry the assumptions are wrong as explain above.

The results are obvious that increasing the number/diameter of Z connection will decrease the swelling.  

NA

Author Response

(The authors gave the same response as above.)

Author Response

Thank you very much for your review.
Please find our response attached.
Sincerely,
Sadegh Mazloomi

Round 2

Reviewer 1 Report

The authors have addressed all reviewer concerns.

Author Response

Thank you very much for your review.
Best regards,
Sadegh Mazloomi

Reviewer 2 Report

Although the authors have made some efforts to address previous comments, they have not adequately addressed a fundamental key element highlighted in the previous review. Consequently, I recommend rejecting this paper.

One of the critical concerns remains the oversimplified moisture movement model, which does not accurately represent the underlying physics. Furthermore, the cited additional papers are of low quality. It is essential to address the model concerning moisture movement during drying based on fundamental principles of heat and mass transfer, as suggested by the reviewer. This aspect has not been adequately addressed.

For instance, the following papers provide examples of the kind of research that should be considered:

·         "A mesoscopic drying model applied to the growth rings of softwood: mesh generation and simulation results"

·         "A relevant and robust vacuum-drying model applied to hardwoods"

·         "A heterogeneous three-dimensional computational model for wood drying"

·         "Finite element analysis of stress-related degrade during drying of Corymbia citriodora and Eucalyptus obliqua"

Moreover, the reviewer requested moisture content validation data in the previous review, which the authors have not provided. Without proper moisture content prediction, the hygroscopic swelling model could yield inaccurate results.

Another concern is the mismatch between the diffusion values in the literature and those cited in the paper, particularly regarding spruce. This inconsistency needs to be addressed for the paper to maintain credibility.

Lastly, there is no additional information added regarding the hygroscopic swelling model, as previously suggested in comment 2. These issues collectively warrant reconsideration and revision before this paper can be accepted for publication.

Author Response

(The authors gave the same response as above.)

Reviewer 3 Report

No further comments

Round 3

Reviewer 2 Report

NA